# Graph Neural Network for Senior High Student's Grade Prediction

Yang Yu , Jinfu Fan, Yuanqing Xian and Zhongjie Wang *

College of Electronic and Information Engineering, Tongji University, Shanghai 201804, China;
1510471@tongji.edu.cn (Y.Y.); fanjinfu@tongji.edu.cn (J.F.); xianyuanqing@tongji.edu.cn (Y.X.)
* Correspondence: wang_zhongjie@tongji.edu.cn

**Abstract:** Senior high school education (SHSE) forms a connecting link between the preceding junior high school education and the following college education. Through SHSE, a student not only completes k-12 education, but also lays a foundation for subsequent higher education. The grade of the student in SHSE plays a critical role in college application and admission. Therefore, utilizing the grade of the student as an indicator is a reasonable method to instruct and ensure the effect of SHSE. However, due to the complexity and nonlinearity of the grade prediction problem, it is hard to predict the grade accurately. In this paper, a novel grade prediction model aiming to handle the complexity and nonlinearity is proposed to accurately predict the grade of the senior high student. To deal with the complexity, a graph structure is employed to represent the students' grades in all subjects. To handle the nonlinearity, the multi-layer perceptron (MLP) is used to learn (or fit) the inner relation of the subject grades. The proposed grade prediction model based on graph neural network is tested on the dataset of Ningbo Xiaoshi High School. The results show that the proposed method performs well in the prediction of senior high school student grades.

**Keywords:** senior high school education; artificial intelligence; graph neural network; grade prediction

## 1. Introduction

Senior high school education (SHSE) has been considered as one of the most important education stages. SHSE concludes the k-12 education and also lays a foundation for the subsequent college education. High-quality SHSE should ensure the student processes corresponding knowledge well so that the graduate is more capable of integrating into society and entering into college education.

When a senior high school graduate applies to college, his (or her) grade plays a critical role. In the United States, most colleges require the applicant to offer a grade point average (GPA) of senior high school, the American College Test (ACT), or the Scholastic Assessment Test (SAT). The grade accounts for a large proportion of college admission. In China, by the same token, college admission is mostly based on the grade of China's College Entrance Examination (CEE) [1]. Nearly every Chinese senior high graduate must take the CEE, and the CEE grade is the only criterion for college admission. Most other countries also take a similar policy that considers the grade as the main factor in college admission.

Therefore, it is reasonable to utilize the grade of the senior high student as an indicator to instruct and ensure the effect of SHSE. This paper presents an artificial intelligence grade prediction model, in order to help the student and his (or her) teacher analyze and improve the study performance and teaching strategy. The proposed method utilizes the previous grades of the student to predict the grades in the subsequent examination. Based on the predicted grades, an adaptive study or teaching adjustment (like personalized feedback) can be made by the student or teacher themselves in, or even ahead of, time to ensure good education quality, before the grade decreases. When an examination has been carried out, the grades in the next examination can be predicted. Then, the predicted grades are utilized

to instruct and ensure the effect of education. A flow chart of this process is presented in Figure 1.

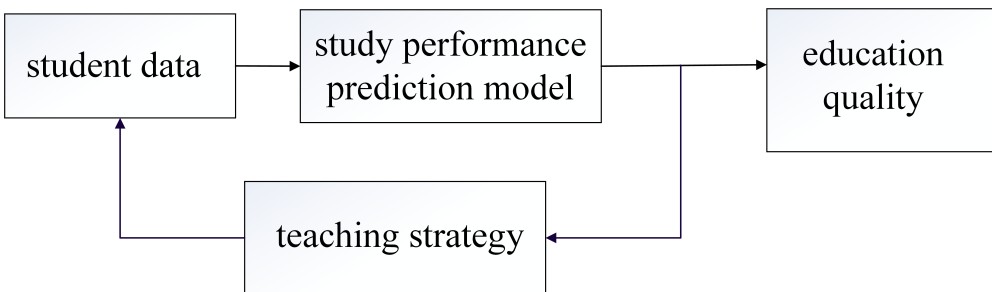

**Figure 1.** The flow chart to ensure the SHSE based on the student performance prediction.

Due to the complexity, nonlinearity, and unclear mechanism of the grade prediction problem, we propose a novel grade prediction method. Firstly, the grade data are represented by the graph structure. Then the multi-layer perceptron (MLP) is employed to learn (or fit) the relation of the graph structure data, to build the grade prediction model. The main idea is to extend the deep learning method applied in 1-dimension or 2-dimension data to graph structure data [2–4]. It is capable of learning (or fitting) the inner relationship among the data in the graph structure [5]. That makes the proposed model suitable for the prediction of senior high school students' grades (seen in Section 2).

Essentially, the proposed model is a kind of supervised learning model [6]. A dataset is needed to train the neural network. In this paper, the dataset of senior high students in Ningbo Xiaoshi High School is utilized. The dataset contains every subject statistic (grade and rank) for the all students in each examination. The dataset and the personalized grade prediction model are both uploaded to the IEEE Dataport (URL: https://ieee-dataport.org/open-access/senior-high-student-dataset-ningbo-xiaoshi-high-school, accessed on 1 October 2021).

### 1.1. Related Work

Since the mechanism of student performance prediction is not clear, most related studies are data-driven, and employ the education data mining (EDM) technology as the analysis tool [7–9]. The student performance prediction problem is related to two factors: data attributes and prediction methods [10]. Data attributes could be various, covering from the subject grade and demographics to dispositions [11,12]. Prediction methods are mainly based on the EDM technologies, which contain linear regression [13], Naive Bayes algorithm [14], support vector machine [15], decision trees [16], and artificial neural networks [17].

Educational research has found that some specific data attributes (such as student's grades and attendance) are particularly relevant to predicting student performance. Bowers et al. [18] systematically analyze the effect of some specific attributes of student data utilized in predicting future student outcomes. These attributes are combined to generate a rule-based prediction model. For instance, a student should be predicted with low outcomes if he (or she) has either low grades or low attendance rates. Similarly, T. M. Christian and M. Ayub [19] use the NBTree classifier, which combines the decision tree classifier and the Naive Bayes classifier, to create a set of rules to predict the student performance. R. Luzcando et al. [20] build a collective student model to predict the student behavior in a procedural training environment. The point is that the collective student model is built based on the clustered past student action logs in the procedural training class. That makes the proposed collective model essentially a rule set-based model. A major drawback of such rule-based prediction models is the poor generalization ability, which means the models might work well for these specific situations (which satisfy the rule set well), but result in poor performance when the situation is not met.

Recent research in data mining addresses the limitations of such rule set-based models by advocating automated learning methods. The regression method is employed in research [11,12,21]. In [11], the student engagement (data attribute) is included to build a regression-based predictor for undergraduate GPA prediction. A. Pardo et al. [12] combine the university student's self-regulated learning indicators and engagement with online learning logs to predict the student's academic performance. R. Conijn et al. [21] utilize the data collected from the Moodle learning management system to predict and compare the student performance of 17 blended courses by a multiple linear regression method. S. Kotsiantis et al. [22] utilize the Naive Bayes algorithm to predict the performances of students (fail or pass) in the final examination of the university-level distance learning, with the student's demographic characteristics and marks on a few written assignments. In [23], five data mining methods (random forest, AdaBoost, logistic regression, support vector machine, and decision trees) are inspected in the senior high student outcome prediction of two U.S. school districts. Lykourentzou et al. [24] propose a combined method to predict the dropout rates of students in e-learning courses (computer networks and communications, web design). Student logs extracted from the learning management system that hosts the e-learning courses are utilized. The prediction model is composed of three methods (feed-forward neural network, support vector machine, and probabilistic ensemble simplified fuzzy ARTMAP) in a voting mechanism. If one of the three methods votes for the dropout prediction, the student is predicted as dropout-prone.

With the booming development of artificial intelligence [25], the artificial neural network is employed in education analysis and prediction [26], due to its capacity of learning (or fitting) the nonlinear relation. B. Guo et al. [27] employ the multi-layer perceptron (MLP) to predict the student's academic performance. The unsupervised learning algorithm, named sparse auto-encoder, is used to pre-train the MLP. Then, the back-propagation method is utilized to fine-tune the MLP. J. Xu et al. [28] develop a novel machine learning method for predicting student performance in degree programs, considering the student's evolving progress. The machine learning method contains the ensemble-based progressive prediction part (to predict the performance) and the course relevance discovering part (to consider students' evolving progress). In the first part, a bi-layered structure comprising multiple base predictors and a cascade of ensemble predictors is applied. In the second part, a data-driven approach based on latent factor models and probabilistic matrix factorization is employed. In [29], MLP is utilized to predict the career strand of the senior high school student (such as general academic strand, sciences, technology, engineering, and mathematics strand, accountancy, business, and management strand). The data of 293 students in 11th grade are considered. K. T. Chui et al. [30] use a deep support vector machine to predict student performances with school and family tutoring. H. A. Mengash [31] predicts the applicant's academic performance to support decision-making in the university admission system. Four methods (artificial neural network, decision tree, support vector machine, and Naive Bayes) are compared, and the artificial neural network outperforms. As a summary, the recent works are presented in Table 1.

The analysis of the literature reveals that the studies of student performance prediction and analysis evolve from the rule set-based predicting method to the data mining method and the artificial intelligence method, due to the more powerful feature extracting and representing abilities provided by the artificial intelligence method. These abilities make the artificial intelligence method more capable of learning (or fitting) the nonlinear model. As described in [31], the artificial neural network outperforms the other methods (decision tree, support vector machine, and Naive Bayes) in the academic performance prediction.

However, most related published studies that address the performance prediction problem focus on building a universal model which applies to all students. For instance, H. A. Mengash [31] considers a dataset that contains 2039 students' information, and uses the artificial neural network to build a unified model to predict each student's academic performance, in order to give some suggestions for the university admission system. This leads to a drawback that the individual features, such as the specific learning style,

variation tendency with time, etc., are not considered, which might reduce the prediction performance. Besides, although most neural network based student performance prediction models achieve relatively good performances, they only treat the model as an end-to-end learning process without investigating the relationship between the data, causing the models to lack interoperability.

**Table 1.** Recent Studies on Student Performance Prediction.

| Reference | Data Attributes | | | Methods | | | Model Type |
|---|---|---|---|---|---|---|---|
| | Demographic | Assignment | Grade | RSM | TLM | DLM | |
| [11] | | | ✓ | | ✓ | | Universal |
| [12] | | ✓ | ✓ | | ✓ | | Universal |
| [18] | ✓ | | ✓ | ✓ | | | Universal |
| [19] | ✓ | | ✓ | ✓ | | | Universal |
| [20] | | ✓ | | ✓ | | | Universal |
| [21] | | ✓ | ✓ | | ✓ | | Universal |
| [22] | ✓ | ✓ | | | ✓ | | Universal |
| [23] | ✓ | | ✓ | | ✓ | | Universal |
| [24] | ✓ | ✓ | | | ✓ | | Universal |
| [27] | ✓ | ✓ | ✓ | | | ✓ | Universal |
| [28] | | ✓ | ✓ | | | ✓ | Universal |
| [29] | | | ✓ | | | ✓ | Universal |
| [30] | | ✓ | ✓ | | | ✓ | Universal |
| [31] | | | ✓ | | | ✓ | Universal |
| Proposed method | | | ✓ | Graph structure data and MLP | | | Personalized |

### 1.2. Contribution

To deal with the above problems, a novel personalized grade prediction model of senior high students is proposed. Different from the above published studies, the individual differences are considered by utilizing the personal data in time series. Moreover, considering the complexity and linearity, the personal data are represented in a graph structure, and the multi-layer perceptron (MLP) is employed to learn (or fit) the inner relation of the graph structure data. To the best of our knowledge, this is the first time that graph structure data and MLP are used in student performance prediction.

The rest of this paper is organized as follows. Section 2 presents the proposed grade prediction method. Section 3 shows the experiment results. Section 4 discusses the application of deep learning methods in the education field. Finally, Section 5 concludes this paper.

## 2. Method

Our purpose is to predict the personalized grades of senior high students accurately dealing with the challenge of complexity, nonlinearity, and unclear mechanism of the grade prediction problem. An artificial intelligence student grade prediction model based on graph structure data and MLP is proposed to satisfy the prediction accuracy requirement.

### 2.1. Graph Neural Network

By combining the graph structure data with the neural network, the named graph neural network can be obtained. The main idea of the graph neural network is to extend the powerful learning (or fitting) ability of the neural network in 1D sequence data (natural language) and 2D graphical data (image) to the graph structure data.

Graph Network Block

The first is to represent the data in a graph structure. The graph neural block, as the unit of the graph neural network, is defined as a graph $G = (\mathbf{u}, V, E)$ to represent the data in the graph structure. $\mathbf{u}$ represents the global attribute. $V = \{\mathbf{v}_i\}_{i=1:N^v}$ is the set of nodes

(of cardinality $N^v$), where $\mathbf{v}_i$ represents the attribute of node $i$. $E = \{(\mathbf{e}_k, r_k, s_k)\}_{k=1:N^e}$ is the set of edges (of cardinality $N^e$), where $\mathbf{e}_k$ represents the attribute of edge $k$. $r_k$ represents the index of the receiver node. $s_k$ represents the index of the sender node. Figure 2 shows the graph network block.

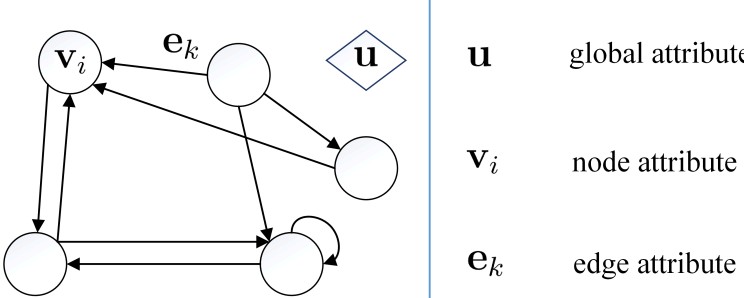

**Figure 2.** Graph network block (used to represent the data in the graph structure).

The second is to apply the neural network in the graph structure data. To do the computation in the graph structure data, three "update" functions $\phi$ and three "aggregation" functions $\rho$ are defined in the graph network block. The neural networks are employed to learn (or fit) these functions.

$$
\begin{aligned}
\mathbf{e}'_k &= \phi^e(\mathbf{e}_k, \mathbf{v}_{r_k}, \mathbf{v}_{s_k}, \mathbf{u}) & \bar{\mathbf{e}}'_i &= \rho^{e \to v}(E'_i) \\
\mathbf{v}'_i &= \phi^v(\bar{\mathbf{e}}'_i, \mathbf{v}_i, \mathbf{u}) & \bar{\mathbf{e}}' &= \rho^{e \to u}(E') \\
\mathbf{u}' &= \phi^u(\bar{\mathbf{e}}', \bar{\mathbf{v}}', \mathbf{u}) & \bar{\mathbf{v}}' &= \rho^{v \to u}(V')
\end{aligned}
\tag{1}
$$

where $E'_i = \{(\mathbf{e}'_k, r_k, s_k)\}_{r_k=i, k=1:N^e}$, $V' = \{\mathbf{v}'_i\}_{i=1:N^v}$, and $E' = \cup_i E'_i = \{(\mathbf{e}'_k, r_k, s_k)\}_{k=1:N^e}$. The "update" functions $\phi^e$, $\phi^v$, $\phi^u$ are mapped across all edge, node, and global attributes to update the corresponding attributes, respectively. The "aggregation" functions $\rho^{e \to v}$, $\rho^{e \to u}$, $\rho^{v \to u}$ are used to reduce the effect of the set input (i.e., $E'_i$, $E'$, $V'$) into a single element effect that represents an aggregated information. The steps of computation in the graph network block is shown in Algorithm 1 and Figure 3.

The "update" functions $\phi^e$, $\phi^v$, $\phi^u$ and "aggregation" functions $\rho^{e \to v}$, $\rho^{e \to u}$, $\rho^{v \to u}$ can be learned (or fitted) by the neural network, such as the multi-layer perceptron (MLP) neural network, given the function fitting problem is what the artificial neural network good at [17].

---

**Algorithm 1** Computation steps of graph network block.

---

1: **input**: Graph $(E, V, \mathbf{u})$
2: **for** $k \in \{1 \ldots N^e\}$ **do**
3:     $\mathbf{e}'_k \leftarrow \phi^e(\mathbf{e}_k, \mathbf{v}_{r_k}, \mathbf{v}_{s_k}, \mathbf{u})$      # Compute updated edge attributes
4: **end for**
5: **for** $i \in \{1 \ldots N^v\}$ **do**
6:     let $E'_i = \{(\mathbf{e}'_k, r_k, s_k)\}_{r_k=i, k=1:N^e}$
7:     $\mathbf{e}'_i \leftarrow \rho^{e \to v}(E'_i)$          # Aggregate edge attributes per node
8:     $\mathbf{v}'_i \leftarrow \phi^v(\bar{\mathbf{e}}'_i, \mathbf{v}_i, \mathbf{u})$      # Compute updated node attributes
9: **end for**
10: let $V' = \{\mathbf{v}'\}_{i=1:N^v}$
11: let $E' = \{(\mathbf{e}'_k, r_k, s_k)\}_{k=1:N^e}$
12: $\bar{\mathbf{e}}' \leftarrow \rho^{e \to u}(E')$          # Aggregate edge attributes globally
13: $\bar{\mathbf{v}}' \leftarrow \rho^{v \to u}(V')$          # Aggregate node attributes globally
14: $\mathbf{u}' \leftarrow \phi^u(\bar{\mathbf{e}}', \bar{\mathbf{v}}', \mathbf{u})$          # Compute updated global attribute
15: **return**: update Graph $(E', V', \mathbf{u}')$

---

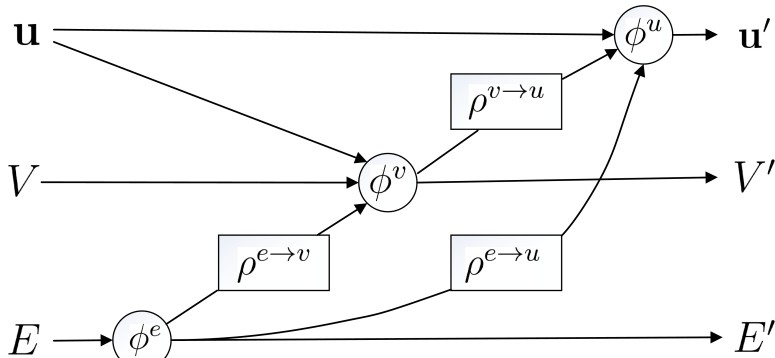

**Figure 3.** Computation flow chart of graph network block.

The third is to construct the graph neural network by composing the graph network block. Because the above graph network block is defined as always taking a graph comprised of edge, node, and global attributes as input, and returning a graph with the same constituent attributes as output, it ensures that the output of one graph network block can be passed as the input to another graph network block. Therefore, the graph network block can be composed to achieve the graph neural network. In the most basic form, two graph network blocks can be composed by passing the output of the first graph network block as the input of the second graph network block. It can be written as $GN_2(GN_1(G))$, where $GN_2$ represents the second graph network block, $GN_1$ represents the first graph network block, and $G$ means the graph structure data. Figure 4 presents the construction of the graph network blocks.

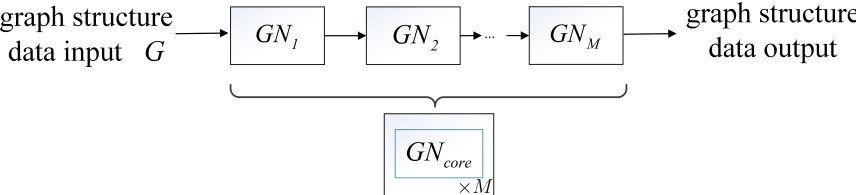

**Figure 4.** Composition of graph network blocks.

### 2.2. Student Grade Prediction Model

Based on the above method, the personalized grade prediction model is able to be achieved. In the model, only the subject grades and ranks of the student are utilized. That is, the student grades and ranks of all subjects in the examination are predicted only based on the previous data.

Firstly, the grades of all subjects are represented in the graph structure, as a graph neural block. Every subject is represented as a node in the graph neural block. The node attribute **v** is the tensor of subject grade and rank. The edge attribute **e** is the tensor of the ratio of the receiver node attribute to the sender node attribute. The global attribute **u** is the tensor of the total grade and average rank. The graph is all connected in bi-direction. Figure 5 shows the graph network block of student grade prediction.

Secondly, set the artificial neural network that are employed to learn (or fit) the "update" functions $\phi^e$, $\phi^v$, $\phi^u$ and "aggregation" functions $\rho^{e \rightarrow v}$, $\rho^{e \rightarrow u}$, $\rho^{v \rightarrow u}$ from the training data. Here, MLP is applied to learn (or fit) the "update" and "aggregation" functions.

Finally, compose the graph network blocks to build the graph neural network which is used for the personalized grade prediction. We compose the graph network blocks in an encode–process–decode architecture, as shown in Figure 6. $GN_{encoder}$ represents the encode graph network block which encodes the input data to graph structure data. Similarly, $GN_{decoder}$ means the decode graph network block which decodes the output graph data (of $GN_{core}$) to student grade data. $GN_{core}$ is formed by composing multiple graph network

blocks (number of $M$) in sequence. $t$ represents the sequence index, i.e., the examination index. $Hidden(t)$ is the output of $GN_{core}$ in $t$ iteration.

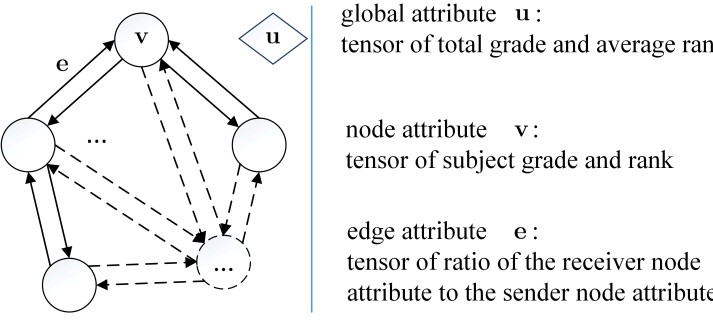

**Figure 5.** Graph network block of student grade prediction.

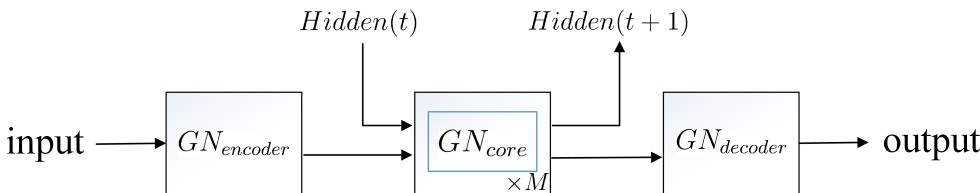

**Figure 6.** Encode–process–decode architecture.

The complete built grade prediction model based on the graph neural network is shown in Figure 7.

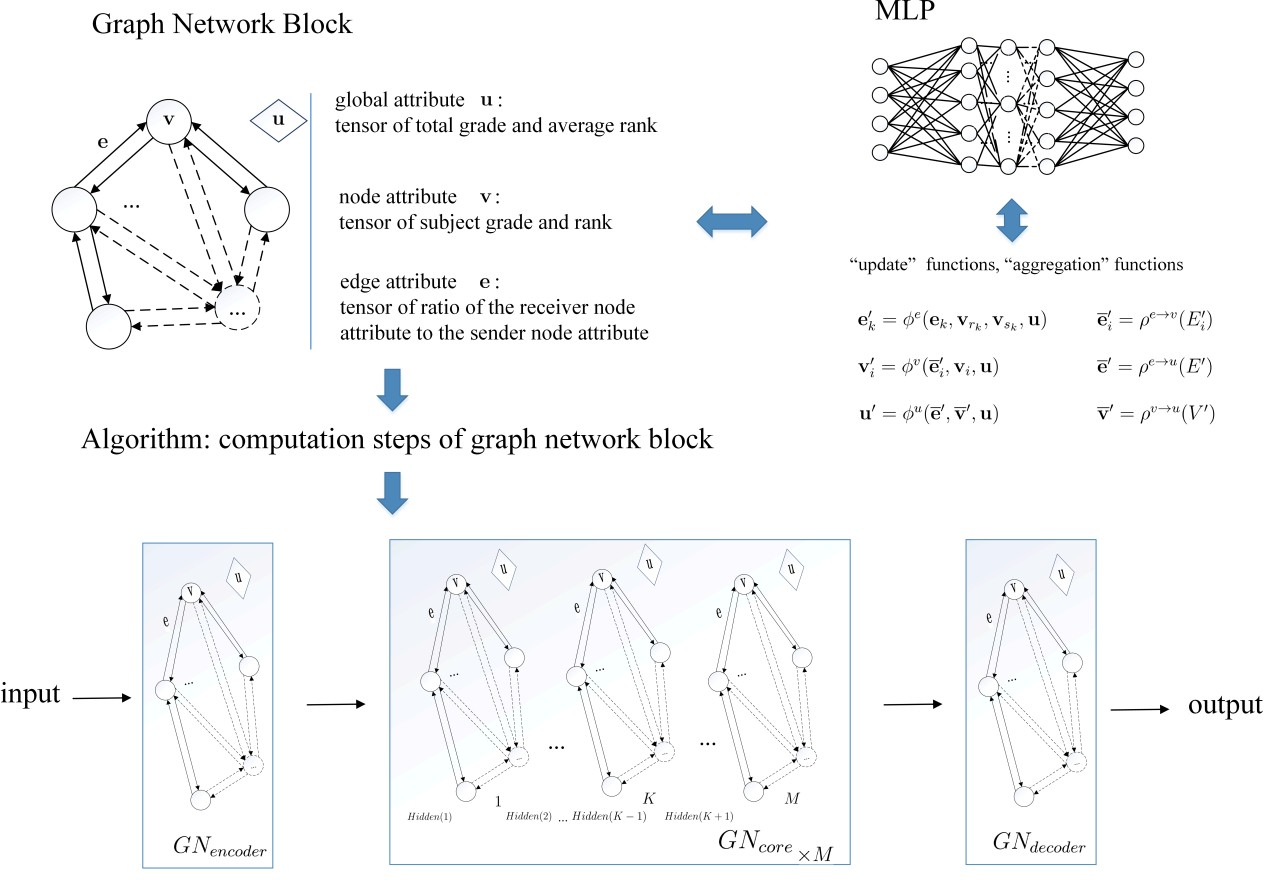

**Figure 7.** Graph neural network model for grade prediction.

### 3. Experiments

The proposed method is conducted in the dataset of senior high students in Ningbo Xiaoshi High School. In order to illustrate the student grade prediction process, we randomly select a student (the student ID is 14295) and build the corresponding personalized grade prediction model. There are nine examination records of the student in the dataset. Each of the examinations contains several subject grades and ranks (in class), seen in Table 2. Due to the limited space, only the math grades and ranks are presented. In fact, the subject grades and ranks contain Chinese, maths, English, physics, chemistry, biology, politics, history, geography, and technology. These subject grades and ranks are shown in Figure 8.

**Table 2.** Examination Records of Student 14295.

| Examination | Date | Maths Grade & Rank |
|---|---|---|
| second semester mid-term exam in 2016SY | 2017/04/20 | 79 & 24 |
| second semester final exam in 2016SY | 2017/06/28 | 77 & 33 |
| first semester mid-term exam in 2017SY | 2017/11/15 | 93 & 02 |
| first semester final exam in 2017SY | 2018/01/26 | null |
| second semester mid-term exam in 2017SY | 2018/04/23 | null |
| second semester final exam in 2017SY | 2018/06/24 | 87 & 10 |
| first semester joint exam with 10 schools in 2018SY | 2018/09/20 | null |
| first semester joint exam with 5 schools in 2018SY | 2018/10/11 | null |
| first semester mid-term exam in 2018SY | 2018/11/05 | 86 & 13 |

SY represents School Year. "Null" means there is no record. Only the math grade & rank is presented to save space.

It can be seen that some subject grades and ranks in some examinations are not recorded. That cannot be used to train or test the proposed grade prediction model. Therefore, we utilize the linear interpolation method [32] to complement the "null" data. The result can be seen in Table 3. Tables 2 and 3 are visualized and compared, as shown in Figure 8. From Figure 8a,b, it can be seen that some subject grades and ranks of the examination (i.e., the first semester final exam in the 2017 school year) are missing. In Figure 8c,d, the missing grades and ranks have been complemented. The dashed line of the grades and ranks of maths in Figure 8c,d illustrates the nonlinearity of the student grades and ranks.

**Table 3.** Examination Data of Student 14295 Complemented by Linear Interpolation Method.

| Examination | Date | Maths Grade & Rank |
|---|---|---|
| second semester mid-term exam in 2016SY | 2017/04/20 | 79 & 24 |
| second semester final exam in 2016SY | 2017/06/28 | 77 & 33 |
| first semester mid-term exam in 2017SY | 2017/11/15 | 93 & 02 |
| first semester final exam in 2017SY | 2018/01/26 | **91 & 04** |
| second semester mid-term exam in 2017SY | 2018/04/23 | **89 & 07** |
| second semester final exam in 2017SY | 2018/06/24 | 87 & 10 |
| first semester joint exam with 10 schools in 2018SY | 2018/09/20 | **87 & 11** |
| first semester joint exam with 5 schools in 2018SY | 2018/10/11 | **86 & 12** |
| first semester mid-term exam in 2018SY | 2018/11/05 | 86 & 13 |

Complementary subject grades are in **bold**.

After the examination data has been complemented, the corresponding graph structure data can be represented. The corresponding graph structure data of the subject grades in the second semester mid-term exam in the 2016 school year is illustrated as an example. The node attribute of maths is $[79, 24]$, which is the tensor of maths grade and rank (in class). The edge attribute **e** is the tensor of the ratio of the receiver node attribute to the sender node attribute. The edge attribute is $[79/74, 24/33]$, if the receiver node is maths and the sender node is physics (the node attribute of physics is $[74, 33]$). The global attribute **u** is the tensor of the total grade and average rank (in class), which is $[690, 36]$.



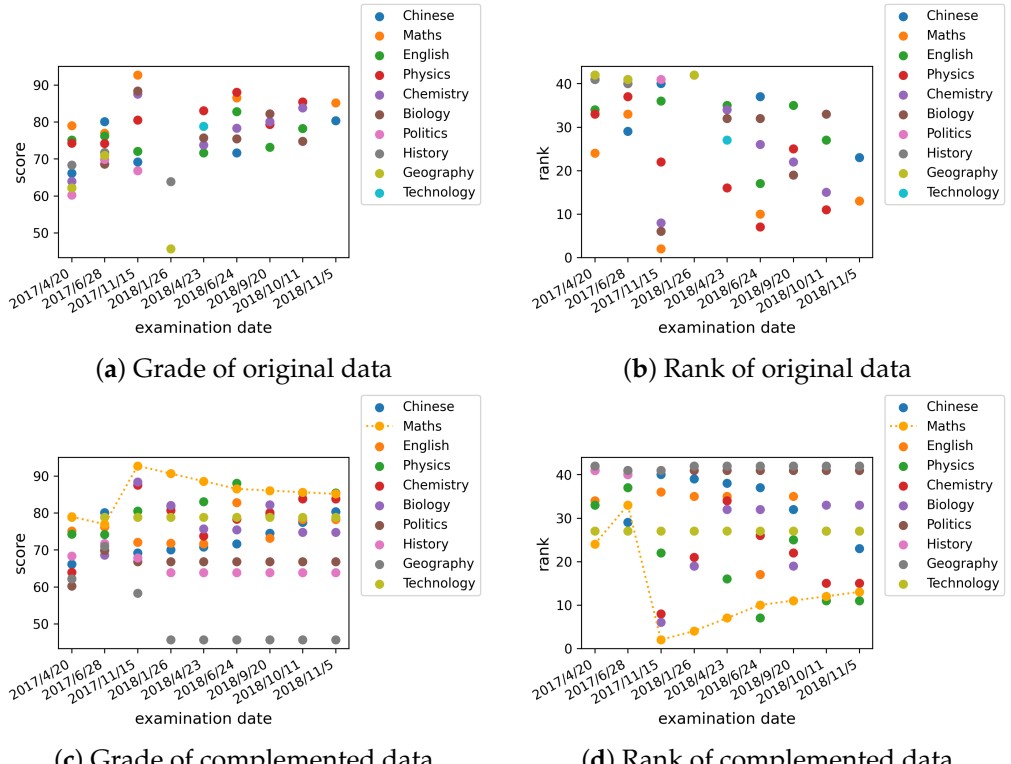

(**a**) Grade of original data          (**b**) Rank of original data

(**c**) Grade of complemented data          (**d**) Rank of complemented data

**Figure 8.** The original data and the complementary data.

With the subject grades and ranks represented in the graph structure, MLP is employed to learn (or fit) the inner relation in the graph structure data. Through extensive trials, the hyperparameters are determined. The MLP contains two hidden layers. Each of the hidden layers has 16 neurons (nodes). The loss function is set to be the mean square error (MSE) [33]:

$$Loss = \frac{1}{n} \sum (||\hat{\mathbf{v}} - \mathbf{v}||^2 + ||\hat{\mathbf{e}} - \mathbf{e}||^2 + ||\hat{\mathbf{u}} - \mathbf{u}||^2) \qquad (2)$$

where $n$ represents the batch size. $\hat{\mathbf{v}}$ means the predicted node attribute (subject grade and rank). $\mathbf{v}$ is the testing node attribute. $\hat{\mathbf{e}}$ means the predicted edge attribute (ratio of the receiver node attribute to sender node attribute). $\mathbf{e}$ is the testing edge attribute. $\hat{\mathbf{u}}$ means the predicted global attribute (total grade and average rank). $\mathbf{u}$ is the testing global attribute. The Adam optimization method [34] is applied to train the model. The learning rate is set to be 0.02.

The performance of the proposed model can then be investigated through the loss function and the error between the predicted grade and the test grade. Firstly, we test the performance of the graph neural network for predicting the grade of student 14295 with the data of nine examinations. The grades of the previous eight examinations are used for training and the grades of the ninth examination are used for testing. The curve of the loss function value shows the performance, seen in Figure 9. Considering that the last value (also the minimal value) of the loss function is 5867.9570, the prediction error (between the predicted grades (of the ninth examination) and the test grades (of the ninth examination)) should be extremely large.

It can be seen that the graph neural network cannot achieve a satisfactory prediction performance in this small data situation. The lack of data leads to the underfitting problem [35].

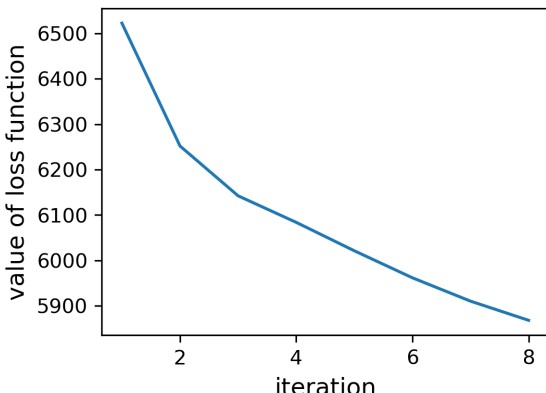

**Figure 9.** The curve of loss function in the data from nine examinations.

In order to deal with the underfitting problem, we use the linear interpolation method to extend the data of student 14295 along the date axis. The extended data (the total size is 565) can be seen partly in Table 4.

**Table 4.** Extended Examination Data of Student 14295.

| Date | Maths Grade & Rank | English Grade & Rank | ... |
|------|--------------------|-----------------------|-----|
| 2017/04/20 | 79 & 24 | 76 & 34 | ... |
| 2017/04/21 | 79 & 24 | 76 & 34 | ... |
| 2017/04/22 | 79 & 24 | 76 & 34 | ... |
| ... | ... | ... | ... |
| 2018/02/18 | 91 & 04 | 72 & 35 | ... |
| ... | ... | ... | ... |
| 2018/11/04 | 86 & 12 | 79 & 27 | ... |
| 2018/11/05 | 86 & 13 | 79 & 27 | ... |

Extended data along the **date** axis.

Then, based on the extended data, the prediction performance of the graph neural network is tested again.

We divide the extended data into three parts, i.e., the training dataset, validation dataset, and testing dataset. The training dataset contains the previous 80% data, which are from 20 April 2017 to 15 July 2018. The validation dataset includes the following 10% data that are from 16 July 2018 to 10 September 2018. The testing dataset contains the last 10% data which are from 11 September 2018 to 5 November 2018. The graph neural network is trained with the training dataset. The loss function curve is shown in Figure 10. The value of the loss function converges and finally reaches 1.2070. The trained model is then used to predict the grades and ranks of the ninth examination. The predicted grades are compared with the real grades (for testing) in Table 5. $\hat{grade}$ denotes the predicted subject grades of student 14295, which is a floating-point type. We ceil the floating-point predicted grade to integer, because the rank and grade are usually integral types. Therefore, the predicted maths grade (85.2500, 86) means the graph neural network's output is 85.2500, then we ceil 85.2500 to 86. *grade* represents the real grade of the ninth examination used for test. The maths grade of the ninth examination is 86, as shown in Table 5. Likewise, the predicted rank and real rank are denoted in the same way.

We can use the relative error as an indicator to inspect the personalized grade prediction performance. For every subject prediction, only the geography grade is predicted with a point error of 1. The real grade is 46, but the predicted grade is 47. The relative error is 1/46. Three subject ranks (maths, chemistry, and geography) all have a rank error of 1. The predicted rank of maths is 12, but the real rank is 13. The predicted rank of chemistry is 16, but the real rank is 15. The predicted rank of geography is 41, but the real rank is 42. The relative error is −1/13 (i.e., (12–13)/13), 1/15, and −1/42, respectively. For the whole

subjects prediction, only one subject (geography) grade and three subject ranks (maths, chemistry, and geography) are not predicted precisely. The relative error of subject grade prediction is 1/10 (total 10 subjects). The relative error of subject rank prediction is 3/10.

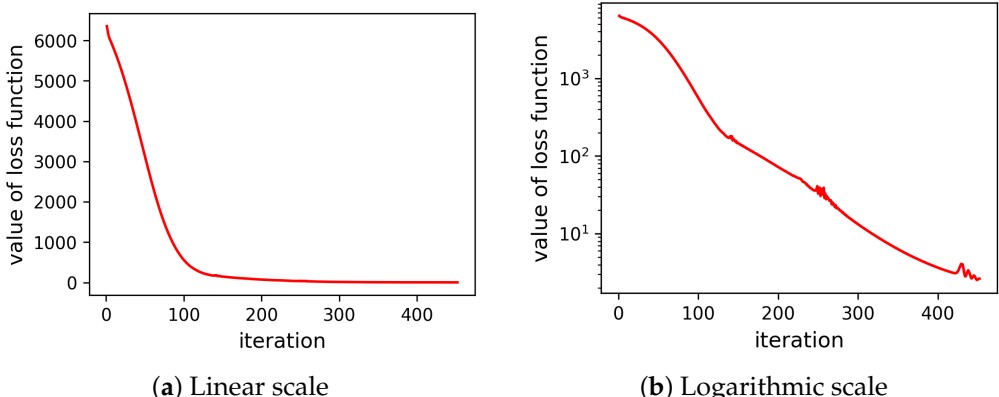

(**a**) Linear scale  (**b**) Logarithmic scale

**Figure 10.** The curve of loss function in extended data.

**Table 5.** Prediction Performance of Proposed Method.

| Subject | $\hat{grade}$(Float, Ceil) | Grade | $\hat{rank}$(Float, Ceil) | Rank |
|---|---|---|---|---|
| Chinese | (80.2998, 81) | 81 | (22.9532, 23) | 23 |
| Maths | (85.2500, 86) | 86 | **(11.8778, 12)** | **13** |
| English | (78.3439, 79) | 79 | (26.9669, 27) | 27 |
| Physics | (85.9123, 86) | 86 | (10.8730, 11) | 11 |
| Chemistry | (83.3443, 84) | 84 | **(15.5037, 16)** | **15** |
| Biology | (74.4902, 75) | 75 | (32.8843, 33) | 33 |
| Politics | (66.8594, 67) | 67 | (40.9425, 41) | 41 |
| History | (63.3881, 64) | 64 | (41.8732, 42) | 42 |
| Geography | **(46.1224, 47)** | **46** | **(40.9427, 41)** | **42** |
| Technology | (78.4384, 79) | 79 | (26.8082, 27) | 27 |

$\hat{grade}$ and $\hat{rank}$ represent the prediction. Grade and Rank represent the test data. Incorrect predictions are in **blod**.

Finally, it can be seen that, with enough data, the graph neural network performs well in the personalized grade prediction. In fact, only the node attribute (tensor of subject grade and rank) is presented in Table 5 to show the performance of the graph neural network based personalized grade prediction model. The global attribute (tensor of total grade and average rank) is also predicted. The predicted total grade is (742.4492 (floating-point), 743 (ceil)), and the real total grade is 743. The predicted average rank is (27.1625 (floating-point), 28 (ceil)), and the real average rank is 27. Similarly, the edge attribute (tensor of the ratio of the receiver node attribute to the sender node attribute, which has 90 elements, so it is not presented without causing confusion) can be easily obtained and compared.

Besides, linear regression (LR) [11,12,21], support vector regression (SVR) [23,31], and multi-layer perceptron (MLP) neural network [27,29,31] are compared to the proposed model. The training dataset, validation dataset, and testing dataset are the same as the proposed model. The corresponding hyperparameters are as follows. For SVR, the radial basis function, which is the most common kernel function, is selected. The MLP contains two hidden layers. Each of the hidden layers has 16 neurons (nodes). The Adam optimization method is employed to train the MLP. The learning rate is set to be 0.02. The mean absolute percentage error (MAPE) is used to evaluate the prediction performances. For the student 14295, the prediction performances of the different methods are compared and presented in Table 6. It can be seen that the proposed prediction model has the best performance for predicting the grades and ranks of the student 14295.

**Table 6.** Prediction Performances (MAPE) of Different Methods for Student 14295.

| Proposed Method | | LR | | SVR | | MLP | |
|---|---|---|---|---|---|---|---|
| grade | rank | grade | rank | grade | rank | grade | rank |
| 0.22% | 1.67% | 9.31% | 44.4% | 7.18% | 36.1% | 4.25% | 23.11% |

Additionally, considering the generality, we extend the experiment to 100 students randomly selected from the dataset of Ningbo Xiaoshi High School, to investigate the performance of the proposed method on a relatively large dataset. The other comparing methods (i.e., LR, SVR, and MLP) are also carried out to compare with the proposed method. For each of the 100 students, the subject grades and ranks are predicted the same as student 14295. Therefore, we obtain in total 100 MAPE items from the 100 students. To summarize the 100 MAPE items and investigate the overall performances, we present the mean of 100 MAPE items and the corresponding standard deviation in Table 7. It can be seen that the proposed method has the best performance.

**Table 7.** Prediction Performances (MAPE) of Different Methods for 100 Students.

| | Proposed Method | | LR | | SVR | | MLP | |
|---|---|---|---|---|---|---|---|---|
| | grade | rank | grade | rank | grade | rank | grade | rank |
| Mean | 0.29% | 10.87% | 2.32% | 43.48% | 4.44% | 60.59% | 2.43% | 43.99% |
| SD | 0.0051 | 0.1164 | 0.0104 | 0.3271 | 0.0147 | 0.4634 | 0.0119 | 0.3763 |

In a nutshell, the experiment results show that the proposed method performs well in the senior high student's personalized grade prediction.

## 4. Discussion

### 4.1. The Application of Deep Learning Method in High School Education

Dealing with the complexity, nonlinearity, and unclear mechanism of the student performance prediction problem, in this study, a kind of deep learning method, i.e., graph neural network is applied to the senior high student's personalized grade prediction and performs well.

However, a satisfying prediction performance is achieved with the student's personal examination data extended by the linear interpolation method. Whereas, the prediction performance degenerates rapidly with the data of only nine examinations. Admittedly, the deep learning method is very good at learning (or fitting) the complex function (mechanism) hidden in the data. However, this ability is based on big data and increasing computation force.

Therefore, the main limitation of deep learning methods applied in education currently is the amount of data. In order to meet the requirement of the amount of data, the linear interpolation method is employed to extend the data, in this study. The reason for employing the linear interpolation method is that we assume students' grade performance would not change rapidly during a short time. The prediction error reflected by the curve of the loss function in Figure 10 converges around 300 iterations and finally reaches 1.2070. That means with the linear interpolation data between the second semester mid-term exam in the 2016 school year (date 20 April 2017), and the second semester mid-term exam in the 2017 school year (date 23 April 2018), which is about 300 data points, the graph neural network based personalized grade prediction model has been trained well. The trained model with about 200 data points should be able to perform well, considering the curve of the loss function shown in Figure 10.

Despite the data being extended by the linear interpolation method, the graph neural network based prediction model eventually performs well in this study. It is still recommended to collect more real data, which could be more diverse and informative, to support the application of the deep learning method in the education field.

*4.2. The Education Analysis Based on the Predicted Result*

With the predicted results (node attribute, edge attribute, and global attribute), a personalized education analysis can be studied. This will be our next work.

The node attribute is the predicted performance of the student in the next examination, which reflects the changing trend in student performance. That can be utilized as feedback to the student or teacher to make a corresponding adjustment and ensure the effect of education in, or even ahead of, time, before education quality decreases.

The global attribute, which represents the total grade and average rank, has a similar function in the education analysis.

The edge attribute is the ratio of two subject grades and ranks (the receiver node attribute to the sender node attribute), which reflects the similarity between two subjects of the student. Along the time axis, the trend in similarity of the specific student can be investigated. Between two different students, the similarity can be compared. The cluster method might be applied to analyze the similarity between the different students. Therefore, the effective learning method or teaching strategy of an individual student can be introduced to the other students belonged to the same cluster. This will be helpful to supply a high-quality education.

**5. Conclusions**

The aim of this study is to support a high-quality senior high school education. Based on the predicted senior high school student's grade, the student and teacher can adjust the study and teaching strategies before the education quality decreases.

Therefore, a graph neural network based senior high students grade prediction model is proposed, dealing with the complexity, non-linearity, and unclear mechanism of the performance prediction problem. The grades and ranks are represented in the graph structure. The node attribute is set to be the tensor of the subject grade and rank. The edge attribute is set to be the tensor of the ratio of the receiver node attribute to the sender node attribute. The global attribute is set to be the tensor of the total grade and average rank. The linear interpolation method is applied to complement and extend the data, dealing with the underfitting problem. The experiment results show that the proposed method is able to precisely predict the grade of the senior high school student.

**Author Contributions:** Conceptualization, Y.Y. and Z.W.; methodology, Y.Y. and J.F.; software, Y.Y. and Y.X.; validation, Y.Y. and Z.W.; writing, Y.Y. All authors have read and agreed to the published version of the manuscript.

**Funding:** This research was funded by National Key R&D Project (2018YFB1702703).

**Institutional Review Board Statement:** Not applicable.

**Informed Consent Statement:** Not applicable.

**Data Availability Statement:** The data can be found in IEEE Dataport. The corresponding URL is https://ieee-dataport.org/open-access/senior-high-student-dataset-ningbo-xiaoshi-high-school , accessed on 1 October 2021.

**Conflicts of Interest:** The authors declare no conflict of interest.

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
