# Peer review of "Graph Neural Network for Senior High Student’s Grade Prediction"

_applsci, doi:10.3390/app12083881_

Round 1
Reviewer 1 Report
The topic addressed in the manuscript is potentially interesting and the manuscript contains some practical meanings, however, there are some issues which should be addressed by the authors:
1) Proposed method should be detailed. Which was the method developed by authors?
2) The authors should clearly emphasize the contribution of the study. Please note that the up-to-date of references will contribute to the up-to-date of your manuscript. The studies named- Coronavirus (covid-19) classification using CT images by machine learning methods
Author Response
Reviewer 1:
The topic addressed in the manuscript is potentially interesting and the manuscript contains some practical meanings, however, there are some issues which should be addressed by the authors:
1) Proposed method should be detailed. Which was the method developed by authors?
[Response 1] We propose a novel student grade prediction method that combines the graph structure represented data with the neural network to deal with the linearity and complexity of the problem to obtain an accurate prediction. Considering the reviewer’s suggestion, firstly, we modify the Abstract to make the proposed method clear (by adding a sentence that describes the proposed method) (from line 6 to line 10 in the manuscript). Secondly, two new paragraphs are added at the end of Section 1 to describe the novelty of this paper (from line 122 to line 141 in the manuscript). Thirdly, linear regression, support vector regression, and the multi-layer perceptron (MLP) neural network are compared with the proposed method in Section 3 (Experiments) (from line 288 to line 309 in the manuscript). All the modifications are marked in red.
2) The authors should clearly emphasize the contribution of the study. Please note that the up-to-date of references will contribute to the up-to-date of your manuscript. The studies named- Coronavirus (covid-19) classification using CT images by machine learning methods.
[Response 2] As the reviewer suggests, we add a new table (table 1 in the manuscript) which makes a review of existing methods and compares linear regression, support vector regression, and MLP with the method proposed in the manuscript (from line 122 to line 141 in the manuscript), to make the contribution of the study clear. Besides, we also cite several up-to-date references including the named studies (i.e., reference [25]).
Finally, we would like to thank the reviewer for the careful reading, helpful comments, and constructive suggestions, which have significantly improved the presentation of our manuscript.

Reviewer 2 Report
- Without data on the size of the training and testing sets, it is not possible to evaluate the correctness of the results.
- Lack of validation data and information about the possibility of overfitting
- Fig. 8 for such large loss function values it is necessary to adopt a logarithmic scale, it is not clear what happens for 200-500 iterations.
- There is no general statement about errors in the testing process - comparing results for 1 sample is not enough.
Author Response
Reviewer 2:
1) Without data on the size of the training and testing sets, it is not possible to evaluate the correctness of the results.
[Response 1] It is really true as the reviewer suggests. However, the reality is that for every student, the dataset only contains 9 examinations data. Firstly, we test the proposed method based on the 9 examinations data (the previous 8 examination data are set to be the training dataset and the last examination data are set to be the testing dataset). But, due to the small size of the dataset, the underfitting problem happens that leads to the poor performance of the proposed method. Therefore, we extend the dataset and test the proposed method again based on the extended dataset. The extended data contains 565 data items. And we divide the extended dataset into three parts i.e., the training dataset, the validation dataset, and the testing dataset. With the training dataset used to train the proposed model, the validation dataset utilized to handle the overfitting problem, and the testing dataset employed to test the performance, the proposed method presents a relatively good performance. The corresponding modifications are marked in red and from line 255 to line 259 in the manuscript.
2) Lack of validation data and information about the possibility of overfitting.
[Response 2] We totally understand the reviewer's concern. Without the validation data, the overfitting cannot be verified. Considering the reality that the original data are insufficient, we extend the data and divide the extended dataset into the training dataset, validation dataset, and testing dataset to investigate the performance of the proposed method. Additionally, some other methods i.e., linear regression, support vector regression, and the multi-layer perceptron (MLP) neural network are compared with the proposed method. The results show the proposed method has the best performance in the testing dataset (consist with the performances in the training dataset and the validation dataset). That shows the proposed method does not have an overfitting problem. The corresponding modifications are marked in red from line 255 to line 259, and from line 288 to line 298 in the manuscript.
3) Fig. 8 for such large loss function values it is necessary to adopt a logarithmic scale, it is not clear what happens for 200-500 iterations.
[Response 3] Thank the reviewer for pointing out this problem in the manuscript. We add the logarithmic scale of Fig 8 (see Fig 10 in the revised manuscript), which shows the value of the loss function reaches 10 around 300 iterations.
4) There is no general statement about errors in the testing process - comparing results for 1 sample is not enough.
[Response 4] It is really true as the reviewer suggests. In order to obtain a general statement about the errors in the testing process and the performance of the proposed method, we expand the testing process to more students (100 students randomly selected from the dataset). The experiments show the good generality of the proposed method. The corresponding modifications are marked in red and from line 299 to line 309 in the manuscript.
Finally, we would like to thank the reviewer for the careful reading, helpful comments, and constructive suggestions, which have significantly improved the presentation of our manuscript.

Reviewer 3 Report
I think the paper is interesting. However, I question the novelty of the study. The authors must clearly show the differences between their study and existing ones (in terms of considered data and type of proposed methods). After clearly confirming the novelty, the authors need to provide much more details about the considered data and analysis to show how their method works. The proposed method could not be assessed well because the lack of many important information.
[Comment 1] Novelty and study background
[Subcomment 1a] The novelty of the paper is not clear. The authors must compare this paper with other existing studies, in terms of the type of considered data and proposed methods, then clearly state the novelty of this study. Please present the comparison in a table.
[Subcomment 1b] To provide clear benefit about the study, I suggest the authors explain more about how this research result could be utilized in reality. How to collect the data, how to conduct the evaluation and modification of teaching methods, etc., to provide better importance about the topic.
[Comment 2] Data
[Subcomment 2a] (Section 1) The authors stated that “The dataset contains students’ demographics (such as age, gender) and every subject statistic (grade and rank) in each examination.” However, I could not see any details about the age, gender, and other data in the model explanation and experiment results.
[Subcomment 2b] The authors must clearly list all dependent and independent variables in their study.
[Subcomment 2c] The authors must clearly define the definition of all data, e.g., how the rank is measured (per class or school).
[Subcomment 2d] The authors need to upload the data in an online repository and share the link in the manuscript to allow (1) better understanding by the readers, and (2) reproducibility by other researchers.
[Comment 3] Proposed method
[Subcomment 3a] (page 5) How do the authors define the sequence of graph network blocks?
[Subcomment 3b] (page 5) The parameters of the model must be tuned well. The authors must explain how they did it, and list the final settings/values for all parameters.
[Subcomment 3c] (page 6) How do the authors ensure appropriate degree consideration of different range of dependent variable values, e.g., is there any applied normalization techniques when calculating the loss?
[Subcomment 3d] The authors stated about “the extended data” in page 8. The authors must provide a complete list of variables considered before and after the extension. In addition to that, the authors need to somehow mark the not existing data (that were interpolated).
[Comment 4] Numerical experiments
Can the authors clearly quantify the quality of the interpolation and the GNN separately? If yes, please report them specifically.
[Comment 5] Applicability
What will be the appropriate time span between assessments (between the runs of the proposed method)? It would be related to the active period of the assessment results.
[Comment 6] Clarity and writing quality
[Subcomment 6a] The authors stated that the problem is nonlinear. They must provide the reasons/proofs.
[Subcomment 6b] When citing a reference at the start of any sentence, the authors must mention the authors’ names.
[Subcomment 6c] When presenting Algorithm 1, please provide some comments beside each important line to explain the process well.
[Subcomment 6d] Please revise the mistyped words, e.g., “…optimize method” (page 6).
[Subcomment 6e] (page 10) The authors stated “The edge attribute is the ratio of two subject grade and rank (the receiver node attribute to the sender node attribute), which reflects the correlation between two subjects of the student.” Can we understand two subjects, e.g., as Chinese and Math? Please be clear by always stating the details of the data types to provide a better understanding.
[Subcomment 6f] (page 10) The authors stated “Between two different students, the correlation can be compared.” It is unclear whether the authors have conducted such calculation or not (it is unclear). Please clearly state whether this is an analysis of the current result or suggestions for next study topics.
[Subcomment 6g] (page 10) The authors wrote “Therefore, the effective learning method or teaching strategy of an individual student can be introduced to the other students belonged to the same cluster.” Please explain how they do it, and using which kind of input data. Have the data been considered in this study or not?

Author Response
Reviewer 3:
I think the paper is interesting. However, I question the novelty of the study. The authors must clearly show the differences between their study and existing ones (in terms of considered data and type of proposed methods). After clearly confirming the novelty, the authors need to provide much more details about the considered data and analysis to show how their method works. The proposed method could not be assessed well because the lack of many important information.
[Comment 1] Novelty and study background
[Subcomment 1a] The novelty of the paper is not clear. The authors must compare this paper with other existing studies, in terms of the type of considered data and proposed methods, then clearly state the novelty of this study. Please present the comparison in a table.
[Response 1a] We propose a novel student grade prediction method that combines the graph structure represented data with the neural network to deal with the linearity and complexity of the problem to obtain an accurate prediction. Considering the reviewer’s suggestion, firstly, we modify the Abstract to make the proposed method clear (by adding a sentence that describes the proposed method) (from line 6 to line 10 in the manuscript). Secondly, we add a new table (table 1 in the manuscript) which makes a review of existing methods. Thirdly, two new paragraphs are added at the end of Section 1 to describe the novelty of this paper (from line 122 to line 141 in the manuscript). Finally, linear regression, support vector regression, and the multi-layer perceptron (MLP) neural network are compared with the proposed method in Section 3 (Experiments) (from line 288 to line 309 in the manuscript). All the modifications are marked in red.
[Subcomment 1b] To provide clear benefit about the study, I suggest the authors explain more about how this research result could be utilized in reality. How to collect the data, how to conduct the evaluation and modification of teaching methods, etc., to provide better importance about the topic.
[Response 1b] As the reviewer suggests, we present a figure (Figure 1) and the corresponding description to provide clear benefits of the study, seen from line 30 to line 36 in the manuscript.
[Comment 2] Data
[Subcomment 2a] (Section 1) The authors stated that “The dataset contains students’ demographics (such as age, gender) and every subject statistic (grade and rank) in each examination.” However, I could not see any details about the age, gender, and other data in the model explanation and experiment results.
[Response 2a] Thank you for your good comment. Actually, the dataset contains students’ information such as, age, gender, consumption information, attendance information, and subject grades, ranks of every examination. However, the proposed grade prediction model only uses the data of the subject grades and ranks. Therefore, in the manuscript, only these data are presented. Considering the reviewer’s concern, we explain which kinds of student information are used in the proposed model. The modifications are marked in red and from line 185 to line 188.
[Subcomment 2b] The authors must clearly list all dependent and independent variables in their study.
[Response 2b] Thank the reviewer for pointing out this problem in the manuscript. We list all the variables (the examination records of the student) used in the proposed method at the beginning of Section 3 (Experiments). The variables are the subject grades and ranks, such as the grade and rank of Maths (79 & 24 shown in Table 2). We assume the subject grades and ranks are all dependent on some relation. Therefore, we represent the subject grades and ranks in the graph structure that all nodes are connected in bi-direction. The variables are listed in Table 2 and the corresponding description is added (from line 213 to line 216 marked in red in the manuscript).
[Subcomment 2c] The authors must clearly define the definition of all data, e.g., how the rank is measured (per class or school).
[Response 2c] Thank the reviewer for pointing out this problem in the manuscript. We clearly define all the data and how they are used in the proposed model. The modifications are marked in red and from line 210 to 216, from line 221 to line 225 in the manuscript.
[Subcomment 2d] The authors need to upload the data in an online repository and share the link in the manuscript to allow (1) better understanding by the readers, and (2) reproducibility by other researchers.
[Response 2d] Thank you for your good comment. We upload the data in an online repository i.e., IEEE Dataport DOI:10.21227/vkz7-zs26.
[Comment 3] Proposed method
[Subcomment 3a] (page 5) How do the authors define the sequence of graph network blocks?
[Response 3a] To clearly state the definition of the sequence of graph network blocks, we add a paragraph to explain the construction of the sequence of graph network blocks. It is marked in red and from line 173 to line 182 in the manuscript.
[Subcomment 3b] (page 5) The parameters of the model must be tuned well. The authors must explain how they did it, and list the final settings/values for all parameters.
[Response 3b] We firstly test the proposed method based on the original data (which contains 9 examinations data, the previous 8 examinations data are set to be the training dataset, and the last examination data are set to be the testing dataset). But, due to the small size of the dataset, the underfitting problem happens that leads to the poor performance of the proposed method. Therefore, we extend the dataset and test the proposed method again based on the extended dataset. The extended data contains 565 data items. And we divide the extended dataset into three parts i.e., the training dataset, the validation dataset, and the testing dataset. Through extensive trials, we determine the hyperparameters of the proposed model based on the training dataset and validation dataset and list the final settings/values for all parameters in the manuscript from line 233 to line 238, and from line 287 to line 298.
[Subcomment 3c] (page 6) How do the authors ensure appropriate degree consideration of different range of dependent variable values, e.g., is there any applied normalization techniques when calculating the loss?
[Response 3c] It is a really good comment and we know the reviewer’s concern. Actually, the normalization techniques are not used in the proposed method and the comparing methods i.e., linear regression, support vector regression, and the multi-layer perceptron (MLP) neural network. For the proposed method, the data are represented in the graph data structure and the neural network is employed to learn (or fit) the relation in the graph structure data. The graph structure data contains the node attributes (grade and rank), edge attributes (the ratio of node attributes), and global attributes (the total grade and average rank). The above attributes are relative. The normalization of every attribute can have an effect on the other attribute. As an illustration, if the node attribute has been normalized (by the min-max normalization method), the edge attribute might not be meaningful, considering the zero value in the normalized node attribute. Besides, the global attribute is also not consistent with the normalization (the total grade equals the sum of normalized grades of the node attributes. And the total grade will exceed rang [0, 1]). However, the MLP can deal with a different range of variables with its good learning (or fitting) ability. And the update functions and aggregation functions used to update graph network blocks also ensure the performance using the different range variables. Additionally, the experiments show that the proposed method performs well in a situation without normalization. For the comparing methods, only the previous data is used to predict the corresponding data. For example, if we want to predict the Maths grade of a certain student in the next examination, only the previous Maths grades of the student are used. Therefore, with only one variable, the normalization methods are not employed.
[Subcomment 3d] The authors stated about “the extended data” in page 8. The authors must provide a complete list of variables considered before and after the extension. In addition to that, the authors need to somehow mark the not existing data (that were interpolated).
[Response 3d] We completely list the variables considered before and after the extension. Actually, only the subject grades and ranks are used in the proposed model. And we make a clear statement in the manuscript (from line 209 to line 225). In addition, we present the subject grades and ranks in tables and figures to illustrate the difference before and after the extension. Before the extension, there are only 9 examination data of the subject grades and ranks. However, after the extension, the data reaches 565 items. It is impossible to present all the extended data in the manuscript, due to the length of the paper. Therefore, we only list some of the extended data in table 3. Besides, in the online repository, the reader can find the whole extended data. And we hope that will make the reader have a better understanding.
[Comment 4] Numerical experiments
Can the authors clearly quantify the quality of the interpolation and the GNN separately? If yes, please report them specifically.
[Response 4] Actually, the linear interpolation method is employed to extend the data to deal with the underfitting problem (caused by the lack of data) of the proposed method. We present the performances of the proposed method before and after the extension of the data to verify the effect of the data extension. So, we are afraid that we cannot clearly quantify the quality of the interpolation and the proposed method separately. But, considering the similarity of the linear interpolation method and the linear regression, we present the performance of the linear regression employed to predict the student grade. The modifications are marked in red and from line 288 to line 307 in the manuscript.
[Comment 5] Applicability
What will be the appropriate time span between assessments (between the runs of the proposed method)? It would be related to the active period of the assessment results.
[Response 5] It is a really good comment. When the grade prediction model has been trained well, the appropriate time span between assessments (between the runs of the proposed method) depends on the frequency of examinations. The proposed method is to prevent the decreasing of the student grade and ensure the education effect. That is to make an adaptive study or teaching adjustment (like personalized feedback) by the student or the teacher themselves in or even ahead of time to ensure good education quality, before the grade decreases, based on the predicted grade. Therefore, when an examination has been carried out, the grades in the next examination can be predicted. And the predicted grades are utilized to instruct and ensure the education effect.
[Comment 6] Clarity and writing quality
[Subcomment 6a] The authors stated that the problem is nonlinear. They must provide the reasons/proofs.
[Response 6a] We present the corresponding Figure 8, which shows the grade prediction problem is nonlinear. The modification is marked in red from line 223 to line 225.
[Subcomment 6b] When citing a reference at the start of any sentence, the authors must mention the authors’ names.
[Response 6b] We carefully read the manuscript and mention the authors’ names when citing a reference at the start of the sentence.
[Subcomment 6c] When presenting Algorithm 1, please provide some comments beside each important line to explain the process well.
[Response 6c] The corresponding comments beside each important line are provided in Algorithm 1. The modification is marked in red.
[Subcomment 6d] Please revise the mistyped words, e.g., “…optimize method” (page 6).
[Response 6d] We carefully read the full manuscript to revise the mistyped words.
[Subcomment 6e] (page 10) The authors stated “The edge attribute is the ratio of two subject grade and rank (the receiver node attribute to the sender node attribute), which reflects the correlation between two subjects of the student.” Can we understand two subjects, e.g., as Chinese and Math? Please be clear by always stating the details of the data types to provide a better understanding.
[Response 6e] Yes. That is the definition of the edge attribute. Additionally, in order to clearly state the node, edge, and global attributes, we illustrate these attributes by representing a certain student's subject grades in the graph structure. The modifications are marked in red and from line 226 to line 233.
[Subcomment 6f] (page 10) The authors stated “Between two different students, the correlation can be compared.” It is unclear whether the authors have conducted such calculation or not (it is unclear). Please clearly state whether this is an analysis of the current result or suggestions for next study topics.
[Response 6f] Actually, we have not conducted the study of the correlation between two different students in this manuscript. It is only an idea for the next study and we believe the correlation between two different students can be used to analyse the learning characteristics of the similar students and to improve the education quality.
[Subcomment 6g] (page 10) The authors wrote “Therefore, the effective learning method or teaching strategy of an individual student can be introduced to the other students belonged to the same cluster.” Please explain how they do it, and using which kind of input data. Have the data been considered in this study or not?
[Response 6g] Through the correlation between the students, we can utilize the cluster method (such as k-means) to profile the students who have the similar learning characteristics. Then, if some teaching methods or strategies are effective to one student in the cluster, the teaching methods or strategies will also be believed to work on other students belonged to the same cluster. However, in this study, the purpose is to build an accurate personalized student grade prediction model. We will do the research in the next study.
Finally, we would like to thank the reviewer for the careful reading, helpful comments, and constructive suggestions, which have significantly improved the presentation of our manuscript.

Round 2
Reviewer 2 Report
I accepts all submitted comments.
Author Response
We would like to thank the reviewer for the careful reading, helpful comments, and constructive suggestions, which have significantly improved the presentation of our manuscript.

Reviewer 3 Report
[Comment 1] Novelty
[Subcomment 1a] The authors mentioned in Section 1.1 that “This leads to a drawback that the individual features such as the specific learning style, the variation tendency with time, etc are not considered.” The authors must clearly compare the previous studies and theirs while listing such different features in Table 1.
[Subcomment 1b] The authors mentioned in Section 1.2 that “To the best of our knowledge, it is the first time that graph structure data and MLP are used in student performance prediction.” Please contrast such novelty in Table 1 by adding a row for this current study.
[Comment 2] Data
If the authors only use grade and rank information, please remove explanations about all other type of data that could cause ambiguity (e.g., demographics data), or mention from the start that only grade and rank data are used. Please recheck and revise the whole manuscript.
[Comment 3] Reference
Please move such doi information from the text into the reference list.
[Comment 4] Writing quality and clarity
[Subcomment 4a] There are some mistakes in usage of capital letters, spaces, etc. Please recheck and correct all writing mistakes in the paper.
[Subcomment 4b] Please add an explanation related to the authors’ previous response about the method’s applicability into the manuscript:
“[Comment 5] Applicability
What will be the appropriate time span between assessments (between the runs of the proposed method)? It would be related to the active period of the assessment results.
[Response 5] It is a really good comment. When the grade prediction model has been trained well, the appropriate time span between assessments (between the runs of the proposed method) depends on the frequency of examinations. The proposed method is to prevent the decreasing of the student grade and ensure the education effect. That is to make an adaptive study or teaching adjustment (like personalized feedback) by the student or the teacher themselves in or even ahead of time to ensure good education quality, before the grade decreases, based on the predicted grade. Therefore, when an examination has been carried out, the grades in the next examination can be predicted. And the predicted grades are utilized to instruct and ensure the education effect.”
[Subcomment 4c] Please remove the correlation term from the manuscript to avoid ambiguity with correlation analysis technique used in statistics.

Author Response
Reviewer 3 :
[Comment 1] Novelty
[Subcomment 1a] The authors mentioned in Section 1.1 that “This leads to a drawback that the individual features such as the specific learning style, the variation tendency with time, etc are not considered.” The authors must clearly compare the previous studies and theirs while listing such different features in Table 1.
[Response 1a] Thank you for your nice suggestion. We present a clear description of the comparison between the previous studies and ours, and list such different features in Table 1. The corresponding modification is marked in red and in the manuscript from line 125 to line 129.
[Subcomment 1b] The authors mentioned in Section 1.2 that “To the best of our knowledge, it is the first time that graph structure data and MLP are used in student performance prediction.” Please contrast such novelty in Table 1 by adding a row for this current study.
[Response 1b] We have added the current study in the last row of Table 1 and contrast such novelty that the graph structure data and MLP are firstly used in the student performance prediction.
[Comment 2] Data
If the authors only use grade and rank information, please remove explanations about all other type of data that could cause ambiguity (e.g., demographics data), or mention from the start that only grade and rank data are used. Please recheck and revise the whole manuscript.
[Response 2] Thank you for your nice suggestion. We have removed the other type of data in the manuscript, in order to disambiguate the data type used in the proposed model.
[Comment 3] Reference
Please move such doi information from the text into the reference list.
[Response 3] We remove the IEEE Dataport DOI information from the text. Instead, we give the URL of the IEEE Dataport in the manuscript.
[Comment 4] Writing quality and clarity
[Subcomment 4a] There are some mistakes in usage of capital letters, spaces, etc. Please recheck and correct all writing mistakes in the paper.
[Response 4a] Thank you for pointing out this problem in the manuscript. We make our best effort to recheck and correct the writing mistakes. We have read the manuscript carefully and used a program named “Grammarly” to check it again. We hope that the mistakes have been corrected completely.
[Subcomment 4b] Please add an explanation related to the authors’ previous response about the method’s applicability into the manuscript:
“[Comment 5] Applicability
What will be the appropriate time span between assessments (between the runs of the proposed method)? It would be related to the active period of the assessment results.
[Response 5] It is a really good comment. When the grade prediction model has been trained well, the appropriate time span between assessments (between the runs of the proposed method) depends on the frequency of examinations. The proposed method is to prevent the decreasing of the student grade and ensure the education effect. That is to make an adaptive study or teaching adjustment (like personalized feedback) by the student or the teacher themselves in or even ahead of time to ensure good education quality, before the grade decreases, based on the predicted grade. Therefore, when an examination has been carried out, the grades in the next examination can be predicted. And the predicted grades are utilized to instruct and ensure the education effect.”
[Response 4b] We have added the corresponding in the manuscript, and the modification is marked in red, from line 33 to line 34, and from line 37 to line 39.
[Subcomment 4c] Please remove the correlation term from the manuscript to avoid ambiguity with correlation analysis technique used in statistics.
[Response 4c] Thank you for pointing out the problem. We have removed the correlation term from the manuscript by using the similarity term to replace it.
Finally, we would like to thank the reviewer for the careful reading, helpful comments, and constructive suggestions, which have significantly improved the presentation of our manuscript.
